# MAC-ReconNet: A Multiple Acquisition Context based Convolutional Neural Network for MR Image Reconstruction using Dynamic Weight Prediction

**Sriprabha Ramanarayanan** [* 1,2]                    SRIPRABHA.R@HTIC.IITM.AC.IN
[1] *Indian Institute of Technology Madras (IITM), India*
[2] *Healthcare Technology Innovation Centre (HTIC), IITM, India*

**Balamurali Murugesan** [*1,2]                    BALAMURALI@HTIC.IITM.AC.IN
**Keerthi Ram** [2]                    KEERTHI@HTIC.IITM.AC.IN
**Mohanasankar Sivaprakasam** [1,2]                    MOHAN@EE.IITM.AC.IN

## Abstract

Convolutional Neural network-based MR reconstruction methods have shown to provide fast and high quality reconstructions. A primary drawback with a CNN-based model is that it lacks flexibility and can effectively operate only for a specific acquisition context limiting practical applicability. By acquisition context, we mean a specific combination of three input settings considered namely, the anatomy under study, undersampling mask pattern and acceleration factor for undersampling. The model could be trained jointly on images combining multiple contexts. However the model does not meet the performance of context specific models nor extensible to contexts unseen at train time. This necessitates a modification to the existing architecture in generating context specific weights so as to incorporate flexibility to multiple contexts. We propose a multiple acquisition context based network, called MAC-ReconNet for MRI reconstruction, flexible to multiple acquisition contexts and generalizable to unseen contexts for applicability in real scenarios. The proposed network has an MRI reconstruction module and a dynamic weight prediction (DWP) module. The DWP module takes the corresponding acquisition context information as input and learns the context-specific weights of the reconstruction module which changes dynamically with context at run time. We show that the proposed approach can handle multiple contexts based on cardiac and brain datasets, Gaussian and Cartesian undersampling patterns and five acceleration factors. The proposed network outperforms the naive jointly trained model and gives competitive results with the context-specific models both quantitatively and qualitatively. We also demonstrate the generalizability of our model by testing on contexts unseen at train time.

**Keywords:** Multiple acquisition contexts, Dynamic weight prediction, MRI reconstruction.

## 1. Introduction

Magnetic Resonance imaging (MRI) offers several benefits of non-invasive acquisition and high soft-tissue contrast but suffers from the inherent slow acquisition. The fundamental challenge in MRI is to mitigate the time-intensiveness of acquisition for the betterment of

---

* Contributed equally

patient comfort. Compressed sensing MRI (CS-MRI) methods have been used to accelerate the acquisition and deployed in clinical environments (Jaspan et al., 2015), however, parameter tweaking and iterative computation of non-linear optimization solvers lead to relatively long reconstruction times (Yang et al., 2018). An emerging active research area for faster and efficient reconstruction is the use of convolutional neural networks (CNN) to learn an offline non-linear mapping between the under-sampled (US) input and fully-sampled (FS) target image. Several existing CNN based methods have shown to provide higher quality reconstruction as compared with the CS-MRI methods (Lundervold and Lundervold, 2019). However, the fundamental challenge in translating these methods to an MRI workstation in real clinical environments is that these methods can effectively operate only for a specific input setting used at train time.

Typically, the weights of CNN are learned using abundant training data. In MR reconstruction, the combination of input settings namely anatomy under study, undersampling pattern and acceleration factor decides the distribution of the training data (Lundervold and Lundervold, 2019). We call each combination of input setting as an acquisition context. In general, deep learning models suffer from covariate shift wherein a model trained using data from a particular distribution tend to perform poorly on data from a different distribution (Li et al., 2018a). Hence, the model must be trained separately for each acquisition context. For instance, two anatomical studies (brain and cardiac), five acceleration factors (2x, 3.3x, 4x, 5x and 8x) and two under-sampling patterns (Cartesian and Gaussian) would generate 20 training contexts. We call each such model, a context-specific model (CSM). Training and storing models specifically for each of these contexts raise demands on time and memory thereby limiting applicability in a real clinical setting.

We consider the problem of MR reconstruction flexible to multiple acquisition contexts using a single network. One naive approach to bring in flexibility is to jointly train the model using a large corpus of training data obtained from various contexts. We call this jointly trained model as joint context model (JCM). The JCM is memory efficient as compared to the CSMs. However, CSMs serve as experts in their respective contexts and hence provide better reconstruction as compared to the JCM. Furthermore, a JCM, trained for a set of contexts (for instance 2x, 3.3x, 4x, 5x and 8x) might not be optimal for unseen contexts (like 4.5x).

The decouple learning framework proposed by (Fan et al., 2019) has been applied for multiple parameterized image operators, wherein the weights of the task-oriented base network are decoupled from the network structure and directly learned by a weight prediction network to suit multiple parameter configurations. The framework has also shown to work for unseen parameters in image operators. In MR image reconstruction, each CSM can be viewed as an image operator parameterized by the acquisition context. The structure of each CSM is the same, however they differ by the set of weights. Weight prediction enables the generation of different set of weights for different contexts including unseen contexts. Hence we have chosen the decouple learning framework for handling multiple contexts in a single network. We demonstrate through extensive experiments, that an MR image reconstruction architecture based on decouple learning framework is able to match the performance of all the trained CSMs and also offer reliable reconstruction on unseen acquisition contexts. We summarise our contributions as follows.

- We propose a multiple acquisition context-based network for MRI reconstruction, called MAC-ReconNet, consisting of a reconstruction module and a dynamic weight prediction (DWP) module. The reconstruction module acts as the base network that performs the intended task of undersampled MRI reconstruction. The DWP module takes as input, the numerically encoded acquisition context vector and learns to predict the convolution layer weights of the reconstruction module that dynamically changes with context at run time.

- We show that the proposed approach can handle multiple contexts involving input settings: 1) anatomy under study: cardiac and brain, 2) undersampling pattern: Cartesian and Gaussian 3) acceleration factors: 2x, 3.3x, 4x, 5x and 8x. The contexts considered for the experimental study are (a) Fixed study while sampling pattern and acceleration factors are varied, (b) Fixed US mask pattern while study and acceleration factors are varied. Results show that the proposed network outperforms the JCM and gives competitive results with the CSMs both quantitatively and qualitatively.

- We also show that our proposed approach can provide better reconstruction for 26 out of 28 unseen acceleration factors as compared to the JCM and equally good performance as compared to the CSMs. Our model for this scenario has been trained for five acceleration factors (2x, 3.3x, 4x, 5x, 8x) with Gaussian undersampling pattern and cardiac MRI as the study of interest.

## 2. Related work

**Deep learning based MRI Reconstruction:** Several CNN-based MR reconstruction methods, ranging from standalone architectures (Wang et al., 2016) (Lee et al., 2017) to deep cascade networks (Schlemper et al., 2017) (Sun et al., 2019), (Huang et al., 2019), (Wu et al., 2018), exist in the literature. Among these, the data driven deep cascaded architectures which map US image to FS image have gained more interest owing to their improved learning capabilities and ability to model complex structures from images (Liang et al., 2019). The deep cascaded architectures consist of alternating CNNs with residual connections and data fidelity (DF) blocks. These models are similar to the unrolled optimization steps in CS-MRI, yielding better reconstruction quality (Diamond et al., 2017).

**Dynamic Weight Prediction:** Several recent works on computer vision explore the idea of introducing more flexibility in the network architecture and learning strategies in various aspects like domain generalization and meta learning (Li et al., 2018b). We focus on weight prediction which is a form of meta-learning strategy in neural networks (Lemke et al., 2015). (Hu et al., 2019) uses a dynamic filter generation network for super-resolution of natural images to support arbitrary scale factors. (Jo et al., 2018) proposed a dynamic filter generation network for video super resolution for capturing spatio-temporal neighborhoods of pixels, to eliminate explicit motion compensation. (De Brabandere et al., 2016) used dynamic filter networks for predicting a sequence of future frames in video.

## 3. Methodology

### 3.1. Problem formulation for deep learning based MRI reconstruction with Dynamic Weight Prediction

Let $x \in C^N$ be the desired image to be reconstructed from undersampled k-space measurements $y \in C^M$, $M << N$, such that $y = F_u x$, where $F_u$ is the undersampled Fourier encoding matrix. For undersampled k-space measurements, this system of equations is under-determined and hence the inversion process is ill-defined. The zero-filled reconstruction $x_u = F_u^H y$ is an aliased image due to sub-Nyquist sampling. A CNN-based MRI reconstruction can be formulated as an optimization problem:

$$\operatorname*{argmin}_{x, W^{CNN}} \quad ||x - CNN(x_u|W^{CNN})||_2^2 + \alpha ||F_u x - y||_2^2 \tag{1}$$

The CNN reconstruction is $x_{CNN} = CNN(x_u|W^{CNN})$, where $CNN$ is the forward mapping from US to FS image, parameterized by the network weights $W^{CNN}$. The acquisition context, $\overrightarrow{\gamma}$, a numerically encoded vector representing a combination of input settings, maps to a set of learned weights $W^{CNN}$. A change in the context vector is reflected in the weights of the CNN block. We represent this relationship by a mapping $W^{CNN} = h(\overrightarrow{\gamma})$ where h could be a linear or a non-linear mapping. In the proposed approach the mapping h is learned by a dynamic weight prediction block $DWP$. The CNN block is a fully convolutional network with n layers. The DWP block takes the context vector $\overrightarrow{\gamma}$ as input and outputs the weights of each layer of CNN for that context.

$$W^{CNN} = (W_1, W_2, ..., W_n) = DWP(\overrightarrow{\gamma}) \tag{2}$$

Here $W^{CNN} = W_1, W_2, ..., W_n$ are the weights of the n layers of the CNN block. We use data fidelity (DF) block in k-space domain after the CNN block to ensure that the CNN reconstruction is consistent with the acquired k-space measurements. The data fidelity operation $f_{df}$ can be expressed as,

$$\hat{x}_{df} = \begin{cases} \hat{x}_{CNN}(k) & k \notin \Omega \\ \frac{\hat{x}_{CNN}(k) + \lambda \hat{x}_u(k)}{1+\lambda} & k \in \Omega \end{cases} \tag{3}$$

Here, $\hat{x}_{CNN} = F_f x_{CNN}$, $\hat{x}_u = F_f x_u$, $\Omega$ is the index set of sampled k-space data, $F_f$ is the Fourier encoding matrix, and $\hat{x}_{df}$ is the corrected k-space and $\lambda \to \infty$. The reconstructed image is obtained by inverse Fourier encoding of $\hat{x}_{df}$, i.e. $x_{df} = F_f^H \hat{x}_{df}$.

The reconstruction module is a cascade of $N_c$ CNN blocks with residual connections in each and k-space data fidelity blocks which can be formulated as,

$$x_{CNN,n} = CNN_n(x_{df,n-1}) + x_{df,n-1} \tag{4}$$
$$x_{df,n} = DF_n(x_{CNN,n}) \tag{5}$$

Here $CNN_n$ and $DF_n$ denote the $n^{th}$ CNN and DF block respectively, $n = 1, 2..N_c$, $x_{df,0} = x_u$ and $x_{rec} = x_{df,N_c}$ is the output of the last DF block.

The DWP module consists of $N_c$ DWP blocks, providing weights to the respective CNN block. The weights of the $n^{th}$ CNN block, $W^{CNN_n}$ is given by,

$$W^{CNN_n} = DWP_n(\overrightarrow{\gamma}) \tag{6}$$

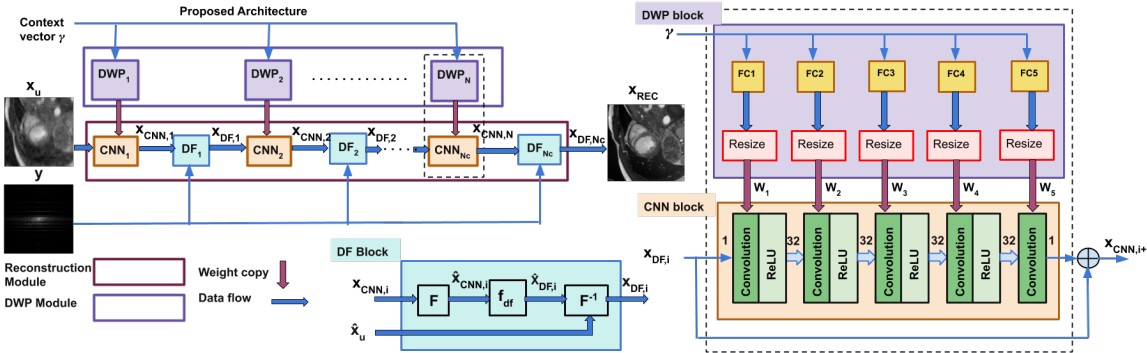

Figure 1: MAC-ReconNet: Proposed architecture for multiple acquisition context based MR Reconstruction. The DWP module takes context vector as input. Reconstruction module takes US image and US k-space as inputs.

### 3.2. Design choices and proposed architecture

**Reconstruction module**: Our design choice for the reconstruction module is the Deep Cascaded Convolution Neural Network (DC-CNN) (Schlemper et al., 2017), one of the state-of-the-art MRI reconstruction network. Other deep cascaded architectures featuring additional computation units like max pooling (Sun et al., 2019), channel attention (Huang et al., 2019) or dense connections (Wu et al., 2018) could still be considered. However, we count on the benefits of DC-CNN from the perspective of simple design yet good quality MRI reconstruction using just the fundamental building units like convolution layers, non-linear activation function and residual connections.

**Dynamic Weight Prediction module**: The core idea of weight prediction is that instead of directly learning the model parameters of a base network, another network could be used to predict the weights of the base network to effectively handle multiple contexts (Ha et al., 2016). The building block of our DWP module is a network with fully connected layers without any non-linear activation units. Using this simple design choice, we explore the possibility of incorporating multiple contexts at the same time meet the performance of CSMs.

**Proposed Architecture:** The proposed architecture (Figure 1) takes the context vector $(\overrightarrow{\gamma})$, the corresponding US image $(x_u)$ and k-space data (y) as inputs and gives the FS image $(x_{rec})$ as output. The architecture has five cascaded functional units. Each functional unit has a CNN block assisted by a DWP block and followed by a DF block. The CNN block has 5 layers and a residual connection wherein the CNN block output is summed with its input. The layers other than the last one has convolution followed by a ReLU while the last layer has only convolution operation. The filter dimensions of convolution layers are given by $(N_{out}, N_{in}, h, w)$, where $N_{out}$ is the number of output channels, $N_{in}$ is the number of input channels and $h \times w$, the kernel size. In our case, the filter dimensions of first and last CNN layer are (32, 1, 3, 3) and (1, 32, 3, 3) respectively and those of the second, third and fourth CNN layer are (32, 32, 3, 3).

The DWP block has 5 fully connected networks FC1 to FC5 corresponding to the weights (filters) $W_1$ to $W_5$ of convolution layers of the CNN block respectively. The number of neurons in each FC network in a DWP block is equal to the size of corresponding convolutional layer weights in the corresponding CNN block. FC1 and FC5 have 288 neurons each corresponding to $W_1$ and $W_5$. Similarly, FC2, FC3 and FC4 each have 9216 neurons each corresponding to $W_2$, $W_3$ and $W_4$.

If $W_i^{FC}$ and $B_i^{FC}$ are the weights and bias values of the $i^{th}$ FC network in a DWP block, the corresponding weight matrix $W_i$ of the $i^{th}$ convolution layer in the CNN block is given by, $W_i = W_i^{FC}\overrightarrow{\gamma} + B_i^{FC}$. Here the size of $W_i^{FC}$ is $(N_w, N_\gamma)$ and $B_i^{FC}$ is $(N_w, 1)$, where $N_w = N_{out} * N_{in} * k * k$ and $N_\gamma$ is size of the context vector $\overrightarrow{\gamma}$ (i.e. 1 or 2). The dimension of $\overrightarrow{\gamma}$ is $N_\gamma \times 1$. During training, the loss calculated between the predicted reconstructed image and the fully sampled target image is back propagated to learn the weights of the DWP block. The weights of the MRI reconstruction network are not made learnable. As a result, the weights of the DWP module are based on context vector input and the image domain loss. The weights $W_i$ are then resized to the actual CNN layer weight sizes and then copied. The five DWP blocks ($DWP_1$ to $DWP_5$) form the DWP module and the five cascaded alternating blocks of CNN ($CNN_1$ to $CNN_5$) and data fidelity ($DF_1$ to $DF_5$) form the reconstruction module.

## 4. Experiment and Results

### 4.1. Dataset and Evaluation metrics

**Dataset Description:** 1) **Cardiac MRI dataset**: Automated Cardiac Diagnosis Challenge (ACDC) (Bernard et al., 2018) consists of 150 and 50 patient records for training and validation respectively. The 2D slices are extracted and cropped to $150\times150$. The number of images for training and validation are 1841 and 1076 respectively. 2) **Brain MRI dataset**: MRBrainS dataset (Mendrik et al., 2015) contains T1, T1-IR and T2-FLAIR volumes of brain for 7 subjects. We use T1 and FLAIR images each with size $240\times240$. For training and validation, 5 subjects with 240 slices and 2 subjects with 96 slices are used. The undersampled images are retrospectively generated for training and testing using Cartesian and Gaussian mask patterns for different acceleration factors (Refer Appendix B). We have limited to real single coil images to demonstrate the ability of the network to multiple contexts. The k-space data used in our simulations is obtained by taking a Fourier transform of the magnitude of the images. **Evaluation metrics**: Peak Signal-to-Noise Ratio (PSNR) and Structural Similarity Index (SSIM) metrics are used to evaluate the reconstruction quality.

### 4.2. Implementation Details

A two stage training process is adopted for the proposed architecture. In the first stage, a functional unit consisting of a $CNN$ and $DWP$ block with a DF block are jointly trained using the L2 loss function. For the training set D consisting of a number of US and FS images as input-target pairs $(x_u, x_t)$,

$$L(\theta) = \sum_{(x_u,x_t)\in D} \|x_t - x_{cnn}\|_2^2 \tag{7}$$

Table 1: Testing on Context with fixed anatomy, varying sampling pattern and acceleration factors. Red denotes best and blue second best performance

| $\overrightarrow{\gamma}$ | | ZF | JCM | MAC-ReconNet (ours) | CSM |
|---|---|---|---|---|---|
| | | PSNR/SSIM | PSNR/SSIM | PSNR/SSIM | PSNR/SSIM |
| Gaussian | 2x | 34.11 ± 2.86 / 0.932 ± 0.02 | 45.37 ± 5.98 / 0.992 ± 0.00 | 46.12 ± 6.82 / 0.994 ± 0.00 | 46.39 ± 6.93 / 0.994 ± 0.00 |
| | 3.3x | 29.2 ± 2.76 / 0.844 ± 0.04 | 40.45 ± 5.01 / 0.98 ± 0.01 | 41.02 ± 5.53 / 0.982 ± 0.01 | 40.99 ± 5.50 / 0.982 ± 0.01 |
| | 4x | 26.96 ± 2.70 / 0.783 ± 0.04 | 38.78 ± 4.62 / 0.972 ± 0.02 | 39.35 ± 5.16 / 0.975 ± 0.02 | 39.14 ± 5.00 / 0.974 ± 0.02 |
| | 5x | 25.56 ± 2.74 / 0.728 ± 0.05 | 37.13 ± 4.27 / 0.961 ± 0.03 | 37.66 ± 4.77 / 0.964 ± 0.03 | 37.35 ± 4.59 / 0.963 ± 0.03 |
| | 8x | 23.30 ± 2.74 / 0.633 ± 0.04 | 33.27 ± 3.78 / 0.918 ± 0.04 | 33.68 ± 3.99 / 0.923 ± 0.04 | 33.42 ± 3.82 / 0.92 ± 0.04 |
| Cartesian | 2x | 29.63 ± 3.17 / 0.843 ± 0.05 | 40.97 ± 4.49 / 0.981 ± 0.01 | 41.64 ± 5.14 / 0.983 ± 0.01 | 41.8 ± 5.37 / 0.983 ± 0.01 |
| | 3.3x | 26.95 ± 3.12 / 0.790 ± 0.06 | 34.81 ± 3.49 / 0.946 ± 0.03 | 34.98 ± 3.54 / 0.948 ± 0.03 | 35.08 ± 3.59 / 0.95 ± 0.03 |
| | 4x | 24.27 ± 3.10 / 0.699 ± 0.08 | 32.79 ± 3.36 / 0.920 ± 0.04 | 33.03 ± 3.36 / 0.923 ± 0.04 | 32.75 ± 3.29 / 0.919 ± 0.04 |
| | 5x | 23.82 ± 3.11 / 0.674 ± 0.08 | 31.79 ± 3.59 / 0.907 ± 0.05 | 32.05 ± 3.47 / 0.909 ± 0.04 | 31.75 ± 3.40 / 0.905 ± 0.05 |
| | 8x | 22.83 ± 3.11 / 0.634 ± 0.09 | 28.53 ± 3.29 / 0.838 ± 0.07 | 28.78 ± 3.21 / 0.842 ± 0.07 | 28.5 ± 3.11 / 0.836 ± 0.07 |

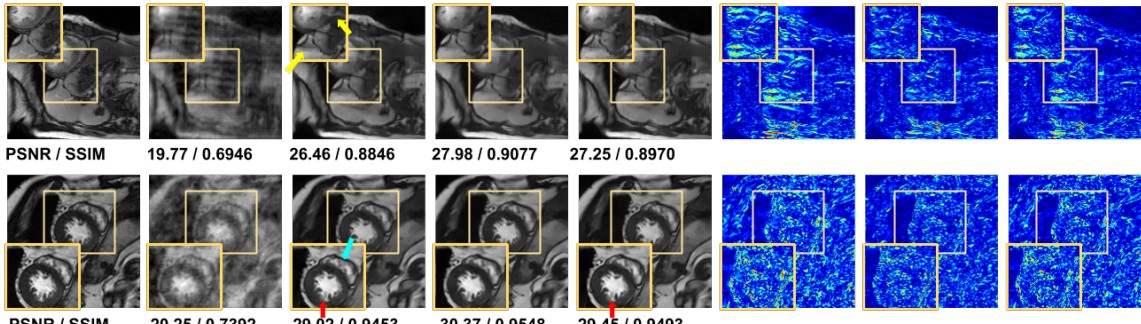

Figure 2: Fixed anatomy, varying sampling pattern and acceleration factors. (Left to right): Ground truth (GT), zero-filled image (ZF), JCM, Ours, CSM, residual image for JCM, Ours and CSM respectively. Top: Cardiac, Cartesian, 5x undersampling. Bottom: Cardiac, Gaussian, 8x undersampling

Here $x_{cnn} = CNN(x_u|W) = CNN(x_u|DWP(\overrightarrow{\gamma}))$ is the predicted image. In the second stage, the weights of this standalone network is used as pretrained weights for the cascaded functional units. The reconstruction and the DWP modules are then trained jointly. In both stages, the models are trained for 150 epochs on Nvidia GTX-1070 GPUs. The models are implemented in PyTorch Adam optimizer is used with a learning rate of 0.001.

### 4.3. Results and Discussion

We evaluate our method on three contexts involving input settings: 1) anatomy under study: cardiac, T1 and FLAIR brain, 2) undersampling pattern: Cartesian and Gaussian 3) acceleration factors: 2x, 3.3x, 4x, 5x and 8x. The contexts considered for the experimental study are (a) Fixed US pattern while study and acceleration factors are varied (b) Fixed study while sampling pattern and acceleration factors are varied. (c) Contexts with acceleration factors unseen at train time.

**Fixed study, varying undersampling pattern and varying acceleration factors:** In this context a combination of mask (Cartesian, Gaussian) pattern for multiple acceleration factors (2x, 3.3x, 4x, 5x and 8x) and for a fixed study i.e cardiac is used. The context is flexible to both scenarios where Cartesian undersampling which is practical to im-

Table 2: Testing on context with Fixed sampling pattern, varying study and acceleration factors. Red and blue indicate the best and the second best performance respectively

| $\overrightarrow{\gamma} : 2 \times 1$ | | ZF | JCM | MAC-ReconNet (ours) | CSM |
|---|---|---|---|---|---|
| | | PSNR/SSIM | PSNR/SSIM | PSNR/SSIM | PSNR/SSIM |
| T1 | 4x | $31.38 \pm 1.02$ / $0.665 \pm 0.02$ | $37.05 \pm 1.44$ / $0.946 \pm 0.00$ | $39.35 \pm 2.04$ / $0.968 \pm 0.00$ | $40.37 \pm 2.09$ / $0.980 \pm 0.00$ |
| | 5x | $29.93 \pm 0.80$ / $0.630 \pm 0.02$ | $35.75 \pm 1.01$ / $0.935 \pm 0.00$ | $38.65 \pm 1.75$ / $0.954 \pm 0.00$ | $39.5 \pm 1.63$ / $0.974 \pm 0.00$ |
| | 8x | $29.93 \pm 0.80$ / $0.630 \pm 0.02$ | $33.47 \pm 1.15$ / $0.905 \pm 0.01$ | $34.3 \pm 0.59$ / $0.907 \pm 0.00$ | $35.21 \pm 1.34$ / $0.939 \pm 0.00$ |
| T2 | 4x | $28.4 \pm 0.84$ / $0.642 \pm 0.02$ | $35.4 \pm 0.09$ / $0.94 \pm 0.00$ | $37.43 \pm 0.37$ / $0.966 \pm 0.00$ | $39.35 \pm 2.04$ / $0.968 \pm 0.00$ |
| | 5x | $26.99 \pm 0.74$ / $0.609 \pm 0.02$ | $33.99 \pm 0.22$ / $0.924 \pm 0.00$ | $37.09 \pm 0.23$ / $0.956 \pm 0.00$ | $37.81 \pm 0.05$ / $0.970 \pm 0.00$ |
| | 8x | $26.49 \pm 0.79$ / $0.588 \pm 0.03$ | $31.7 \pm 0.03$ / $0.899 \pm 0.00$ | $32.68 \pm 0.91$ / $0.912 \pm 0.00$ | $33.35 \pm 0.27$ / $0.93 \pm 0.00$ |

PSNR / SSIM    22.86 / 0.7551    28.57 / 0.9636    31.04 / 0.9756    32.32 / 0.9834

PSNR / SSIM    22.07 / 0.6010    29.25 / 0.9258    33.44 / 0.9713    33.57 / 0.9715

Figure 3: Context with Fixed sampling pattern, varying study and acceleration factors.(Left to right): GT, ZF, JCM, Ours, CSM, residual image for JCM, Ours and CSM respectively. Top: Cartesian, T1 Brain, 5x undersampling: Bottom: Cartesian, Flair brain, 5x undersampling.

plement and simple to reconstruct, is preferred and other kinds of undersampling (Gaussian, spiral or radial) where higher accuracy metrics are preferred (Geethanath et al., 2013). The context vector is a tuple with acceleration factor as the first element and the mask pattern enumerated as 1: Cartesian, 2: Gaussian, as the second element. (For example, context vector [4 2] indicates 4x Gaussian acceleration). We compare our model with 10 respective CSMs and the JCM.

From Table 1, the observations are, 1) The proposed method outperforms the JCM for all the acceleration factors and mask patterns. 2) The proposed approach gives competitive performance as compared to CSMs and for higher acceleration factors (specifically 8x), our model performs better. Our model has the benefit of learning both the common and context specific aspects since it is trained on images with multiple undersampling degradations as compared with the CSMs which are trained on images with fixed degradation. 3) The Gaussian undersampled images exhibit higher PSNR and SSIM metrics than the Cartesian couterparts as intended.

In Figure 2, the regions marked with blue and red arrows in the JCM and CSM show faint and aliased structures, the corresponding regions in our images are more closer to the ground truth. Our method gives least residual errors with respect to the ground truth. The dealiasing effect of our approach (yellow arrow marks in JCM) is more prominent in

Table 3: Testing for unseen contexts. Red and blue indicate the best and the second best performance respectively

| $\vec{\gamma}$ | JCM | MAC-ReconNet (ours) | CSM |
|---|---|---|---|
| | PSNR/SSIM | PSNR/SSIM | PSNR/SSIM |
| 4.8 | 35.57 +/- 3.75 / 0.9493 +/- 0.03 | 36.97 +/- 4.79 / 0.9594 +/- 0.03 | 36.85 +/- 4.46 / 0.9592 +/- 0.03 |
| 5.2 | 34.92 +/- 3.71 / 0.9434 +/- 0.03 | 36.34 +/- 4.64 / 0.9546 +/- 0.03 | 36.37 +/- 4.53 / 0.9541 +/- 0.03 |
| 6 | 33.96 +/- 3.57 / 0.9301 +/- 0.03 | 35.21 +/- 4.32 / 0.9425 +/- 0.04 | 35.06 +/- 4.06 / 0.9418 +/- 0.03 |
| 6.4 | 33.02 +/- 3.58 / 0.9193 +/- 0.04 | 33.99 +/- 4.26 / 0.9321 +/- 0.04 | 34.03 +/- 3.89 / 0.9315 +/- 0.04 |
| 6.8 | 32.68 +/- 3.55 / 0.913 +/- 0.04 | 33.93 +/- 4.21 / 0.9284 +/- 0.04 | 33.98 +/- 3.98 / 0.9277 +/- 0.04 |
| 7.2 | 32.15 +/- 3.60 / 0.904 +/- 0.04 | 33.29 +/- 4.04 / 0.9203 +/- 0.05 | 33.18 +/- 3.72 / 0.9189 +/- 0.04 |
| 7.6 | 31.58 +/- 3.58 / 0.8955 +/- 0.05 | 32.58 +/- 3.93 / 0.9115 +/- 0.05 | 32.55 +/- 3.68 / 0.9102 +/- 0.05 |

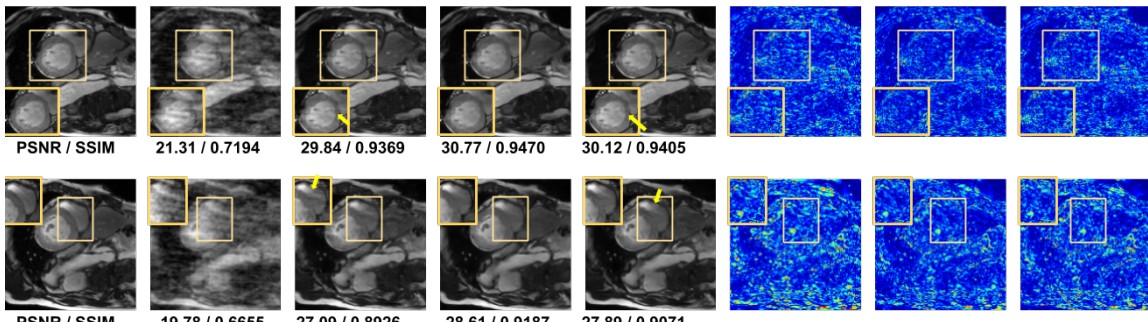

Figure 4: Testing for unseen contexts. Trained model: (Left to right): GT, ZF, JCM, Ours, CSM, residual image for JCM, Ours and CSM respectively.Top: Unseen context 1: Cardiac, Gaussian pattern, 5.2x under-sampling. Bottom: Unseen context 2: Cardiac, Gaussian pattern, 7.6x under-sampling.

the Cartesian case. Preliminary experiments with varying acceleration factors also showed similar performance (Appendix A).

**Fixed under sampling pattern, varying Acceleration Factors and varying studies:** This context demonstrates the flexibility of our model when multiple study sequences of the same anatomy is acquired on the same scanner. One example for such a scenario is the multi-contrast MRI (Liang and Lauterbur, 2002), where in multiple sequences (T1, T2, and proton-density weighted MRI) of the same anatomy are acquired for diagnosis. The context combines multiple studies - T1 and T2 FLAIR brain images with multiple acceleration factors - 4x, 5x and 8x, for a fixed Cartesian mask pattern. The context vector is a tuple with acceleration factor as the first element and the study sequence enumerated as 1: T1, 2: T2 MRI, as the second element. We compare our model with six CSMs and the JCM. From Table 2 we observe that our method consistently outperforms the JCM and approaches the performance of CSMs. Figure 3 shows smudged structures (top blue arrows) and faint structures (bottom red arrow) in the JCM images, our images closely resembles that of the CSM and the target. The residue image is closer to that of the CSM.

**Testing on Unseen Acceleration Factors:** We evaluate our model trained with fixed study (cardiac), fixed undersampling pattern (Gaussian) and varying acceleration factors (2x, 3.3x, 4x, 5x, 8x) on unseen contexts. This experiment gives better insight on the weight generalization behavior of the proposed method on unseen contexts. The

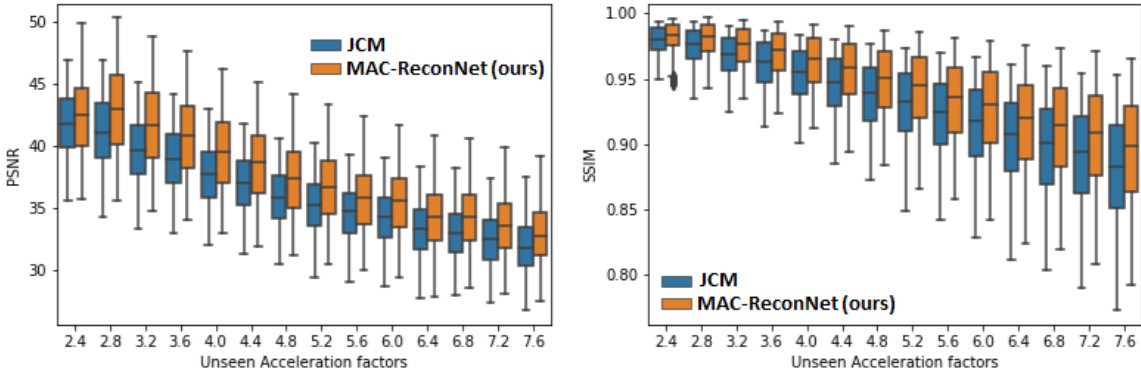

Figure 5: PSNR / SSIM plots for unseen contexts. Left: PSNR metrics vs unseen acceleration factors. Right: SSIM metrics vs unseen acceleration factors. Plots show consistent improvements in metrics of our method as compared with the JCM. For higher acceleration factors, better SSIM values are obtained

context vector has only one element indicating acceleration factor. We create Gaussian undersampled test images with factors varying from 2.4x to 7.6x (not used at train time) in increments of 0.2 making it to 28 unseen contexts. We evaluate our method on all these contexts and compare with the JCM. Training CSMs for all the 28 contexts is cumbersome. So we randomly trained 7 CSMs for 4.8x, 5.2x, 6.0x, 6.4x, 6.8x, 7.2x and 7.6x factors.

Our observations are 1) For 26 out of the 28 unseen contexts, our model outperforms the JCM as shown in the PSNR and SSIM box whisker plots in Figure 5 for 14 of them. 2) Table 3 shows quantitative metrics for 7 unseen contexts. Our model gives equally good metrics as compared to the CSMs and much better metrics than the JCM. Our model and the JCM are trained with the same set of images with just five randomly chosen acceleration factors (2x, 3.3x, 4x, 5x, 8x) covering a wide range from 2x to 8x. The DWP module has no non-linear activation units. Hence the CNN weights and the DWP weights are linearly related thereby providing context-suitable weights for the CNN and meets the performance of the trained CSMs. The JCM, on the other hand, is not generalizable enough for unseen contexts. 3) Figure 4 shows the unseen images for 5.2x and 7.6x undersampling. Figures show that the images are dealiased much better as compared with the JCM or the CSM (region shown with yellow arrow marks in JCM and CSM).

### 4.3.1. Storage efficiency

The number of models to be stored increases linearly with the acquisition contexts. The complexity of storage for existing CNN based MR reconstruction methods is $O(N)$ where N is the number of contexts. For our approach, reconstruction modules are not saved since context-specific weights are provided by the DWP module. Hence storage complexity is $O(1)$ in our case.

### 4.4. Conclusion

In this work, we show that a CNN-based MR reconstruction that exhibits flexibility to multiple acquisition contexts is more appropriate for a clinical scenario. The acquisition contexts are combination of input settings namely anatomy under study, undersampling mask pattern and acceleration factor. The proposed method, called the MAC-ReconNet incorporates flexibility to multiple contexts in a single model, by using a dynamic weight prediction module to generate context-specific weights to our MR reconstruction module. We show that the proposed method performs much better than a model that is jointly trained for multiple contexts and gives competitive results as compared to the context specific models. We also show that the proposed method generalizes well to unseen contexts.

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
