# OpenReview forum: "MAC-ReconNet:  A Multiple Acquisition Context based Convolutional Neural Network for MR Image Reconstruction using Dynamic Weight Prediction"
_MIDL.io/2020/Conference — MIDL 2020_

### Official Review · AnonReviewer3 · 2020-03-12
**A Novel Method for MRI Reconstruction with Extensive Experiments**

**Rating:** 3
**Confidence:** 3
**Recommendation:** Poster

**Summary:**

The authors propose a framework to utilize one model under different acquisition context scenarios.
A novel dynamic weight prediction model is proposed to learn to predict the kernel weights for each convolution based on different context settings.
Experiments show that the proposed method outperforms the model trained on the context-agnostic setting and acquires similar results to models trained by context-specific settings.

**Strengths:**

1).  - The idea of learning convolution weights for different input image quality is novel.
2).  - The method part is well-written and easy to understand.
3).  - It conducts extensive experiments for three different settings and the results demonstrate the effectiveness of the proposed method.

**Weaknesses:**

1).  - Opposite to the Method part, it's hard to read the abstract and introduction. Some typo problems lie here.
2). - It seems that the DWP need to generate a specific weight each time. The authors do not compare the inference speed of the proposed method with others.
3). -  In Table 3., the result of the proposed method is slightly higher than the CSM. There can be more discussion here.

**Detailed Comments:**

1. Some sentences are hard to read, e.g. In the abstract, 'By acquisition context...'.
2. Line 1, 'network based' -> network-based
3. context specific ->  context-specific
4. In the contributions, 'Fixed US mask pattern...', no explanation for 'US'.
5. Same as (4), no explanation for 'FS'.
6. Page 4, 'We represent this relationship by a mapping W = h( γ ) where h could b', h --> $h(\cdot)$.
7. Page 5, '(Ha et al., 2016).' the reference is missing.
8. Page 7, Table 1, 'Red denotes best and blue second best performance' misses the dot.
I strongly recommend the authors to check the typo problem several times.

**Justification Of Rating:**

The authors propose a framework to utilize one model under different acquisition context scenarios. The method is novel with extensive experiments. Results show the effectiveness of the proposed method. But the writing needs to be improved. Therefore I recommend the weak accept.

**Paper Type:**

methodological development

**Questions To Address In The Rebuttal:**

1).  -In Eq.3, what does $\lambda$ mean? The paper does not show an explanation.
2).  -It will be interested to see the model's performance in a harder case: varying anatomy, sampling pattern and acceleration factors at the same time.


**Special Issue:**

no

---

> ### Author Response · Authors · 2020-03-25
> **Reply for three parameter context, Inference speed and changes in problem formulation**
>
> Weaknesses
> ==========
>  1) Analysis on computations and inference speed
>
> Let $t_{mul}$ be the time taken for one floating point (FP) multiplication operation and $t_{add}$ time taken for one FP addition operation and $t_{dc}$ be the time taken for one data consistency unit.
>
> Each CNN block has 5 convolution layers. The first and last layers have 32*3*3=288 weights
> The second, third and fourth layers have 32*32*3*3 = 9216 weights each.
> The reconstruction module has 5 CNN blocks. Hence
> The total number of weights in the reconstruction network is (2*288 + 3*9216)*5 = 2880 + 138240 = 141120 weights.
> Total number of bias values  = number of feature maps in each CNN block *5 = (32+32+32+32)*5 = 128*5 = 640.
> Hence total number of multiplication operations are 141120 * (H * W). where H and W are the height and width of the feature map.
> Total number of addition operations are 640 * H *W.
>
> Time time taken to evaluate one image by the baseline network, DC-CNN , $t_{DC-CNN}$ = (141120 * (H * W) * $t_{mul}$) + (640 * H *W * $t_{add}$) + (5 * $t_{dc}$)
>
> For our network, the DWP module has five DWP blocks in addition to the base network.
> Number of weights in DWP module  =  m * 141120, where m is number of elements in the context vector.
>
> Number of addition operations = 141120
> Time taken by the DWP module  = (m * 141120 *  $t_{mul}$) + (141120  * $t_{add}$)
>
> Total inference time taken by MAC-ReconNet = (141120 * (H * W) * $t_{mul}$) + (640 * H *W * $t_{add}$) + (5 * $t_{dc}$) + (m * 141120 *  $t_{mul}$) + (141120 * $t_{add}$)
> Which reduces to
>
> $t_{DC-CNN}$ +  (m * 141120 *  $t_{mul}$) + (141120 * $t_{add}$)
>
> From this we see that inference speed depends on the size of the context vector.
>
> 2) Results higher than the CSM.
>
> We note that the CSM is the best possible model since it is expertised (well-trained) in its context. At the same, CSMs demand lots of computational burden when more number of acquisition contexts requirements need to be met. Hence bringing the quality of CSM is the objective. Though the reported numbers show improvements over CSM in certain cases, we emphasize that the improvements are random and not significant . However, we believe that similar performance numbers achieved could be due to the fact that by integrating images of multiple contexts in a single network, the network learns more information as compared to the CSMs.
> Questions To Address In The Rebuttal:
>
> Questions To Address In The Rebuttal
> ================================
> 1) 𝜆 in Equation 3, signifies the data consistency weight. Equation 3 uses the predicted image FFT for the non-sampled Fourier frequencies, and fills back the sampled frequencies, along with averaging in the presence of noise. For the noiseless case, 𝜆 →∞, in which case the sampled frequencies are merely restored to their measured values by this operation.
>
> 2) Three parameter context vector experiment
> ===========================================
> We do have the results for this experiment done with the following configuration of acquisition context.
> Context vector - 3 element vector  - [multiple study, multiple mask pattern, multiple acceleration factors].
> Varying anatomy - T1 (enumerated as 1) and T2 Flair (enumerated as 2) Sequences of MRBrainS dataset, enumerated as 1 and 2 respectively)
> Varying Mask pattern - Cartesian and Gaussian mask patterns enumerated as 1 and 2 respectively
> Varying Acceleration factors - 4x, 5x and 8x enumerated as 4, 5 and 8 respectively
> We have reported two of the 12 results as follows.
>
> Table: PSNR/SSIM
> -------------------------
> Context			                     JCM				      Ours  			      CSM
>
> Study  - T1 brain		          36.19 +/- 0.96		   39.49 +/- 2.03		39.5 +/- 1.6
> Mask - Cartesian                    0.9403 +/- 0.01              0.9715 +/- 0.01	        0.9743 +/- 0.00
> Acceleration factor - 5x
> -----------------------------------------------------------------------------------------------------------------------
> Study  - T2 Flair brain		35.43 +/- 0.31		40.58 +/- 0.15		41.06 +/- 0.33
> Mask - Gaussian		        0.9253 +/- 0.00	        0.9771 +/- 0.00	        0.9819 +/- 0.00
> Acceleration factor - 8x
> -----------------------------------------------------------------------------------------------------------------------
> We see that in the three parameter case also, our model performs well and is able
>  to provide quality metrics closer to the corresponding context specific models.
>
> 3) We will make the necessary changes for comments - 1, 2, 3, 4, 5, 6, 8
> Comment 7, the reference is available in the References section.
> It is specified as follows.
>
> David Ha, Andrew Dai, and Quoc V Le. Hypernetworks. arXiv preprint arXiv:1609.09106, 2016.

---

### Official Review · AnonReviewer2 · 2020-03-13
**Interesting paper but lacking of details of methods**

**Rating:** 2
**Confidence:** 4

**Summary:**

This paper proposed MRI reconstruction framework that is flexible to multiple acquisition context and generalizable in real scenarios. To this end, the authors proposed reconstruction module and dynamic weight prediction module which takes acquisition context vector as input. Experimental results support the effectiveness of the proposed method.

**Strengths:**

The authors encoded the combination of input settings (anatomy, undersampling pattern, and acceleration factors) as acquisition context vector, and used it for flexible MRI reconstruction as a single network. This is important since the context-specific model demands lots of computational burden in practice.

**Weaknesses:**

- Lack of details of how to combine DWP block and CNN block in section 3.1.  The notations are very confusing since the network weights W in eq (1) is the weights of CNN block, which is in general independent of acquisition context. Then, the authors used the same “W” for the output of DWP block. Also, the authors are missing period or comma in many places.
- Need more detailed information about dataset such as original dimensions of cardiac data. How are the images cropped?
- Need comparison of model complexity with joint context model. How much is it increased due to DWP module?


**Justification Of Rating:**

The method was not fully explained. The authors proposed dynamic weight prediction block, but it is hard to understand how to encode the acquisition context and how to combine this information with CNN module.

**Paper Type:**

methodological development

**Questions To Address In The Rebuttal:**

- In page 4, how was the acquisition context gamma encoded? How were two weights from DWP block and CNN block combined? In page 6, the authors mentioned “The weights Wi are then resized to the actual CNN layer weight sizes and then copied.” If this is true, then no information from the MR images is used at all.

**Special Issue:**

no

---

> ### Author Response · Authors · 2020-03-25
> **More details on weight update scheme, model complexity, context vector encoding**
>
> Note about Weaknesses
>
> We completely agree with this comment. The usage of notation W in equation 1 is misleading. It gives an impression that the CNN weights are independent of the acquisition context, while actually in the proposed approach, the CNN weights are context dependent, hence we make the following changes so as to bring in clarity in the formulation.
>
> In equation 1, we change the notation W to  $W^{CNN}$ .
> In the paragraph, that follows equation 1, the expression $x_{CNN} = CNN(x_{u}| W)$ is changed as  $x_{CNN} = CNN(x_{u}| W^{CNN})$
>
> The expression, W = h(ℽ) is changed as $W^{CNN} = h(ℽ)$
> Weaknesses
> ============
> 1) The ACDC data has image slices with varying sizes and so to bring the dimensions of all the images same, we have done center cropping of 150x150.
>
> 2) Model complexity generally refers to the layer weights of the network. Each CNN block has 5 CNN layers. Layer 1 and 5 have 288 convolution weights each. Intermediate layers have 9216. Total number of weights amount to  28224 *5 = 141120 and 25 bias values to be stored. These weights are provided by the corresponding DWP layers. Hence model complexity is not increased due to DWP module since it is enough to store the DWP weights alone.
> Another important thing to note here is, in the context-specific model (CSM) scenario, n such CSMs need a storage of n * (141120 + 25) parameters. In our case, its enough to store DWP weights (only once) since DWP dynamically gives weights of each of the n contexts.
>
> Questions To Address In The Rebuttal
> ==================================
> 1) In Section 4.3, Results and Discussion, we have specified the following lines about encoding of the acquisition context vector ℽ,
> The context vector is a tuple with acceleration factor as the first element and the mask pattern enumerated as 1: Cartesian, 2: Gaussian, as the second element.
>
> 2) We do not combine the weights of the CNN and the DWP block. In Section 3.1, equation 6, signifies that weights of the nth CNN block of the reconstruction network is actually given by the output of the nth DWP block.
>
> 3) In our model, the undersampled image is fed to the reconstruction module, the corresponding context vector is fed to the DWP module. During training, the loss calculated between the predicted reconstructed image and the fully sampled target image is back propagated through the reconstruction module to the DWP layers and the weights of the DWP block are learned. The weights of the CNN layers of the MRI reconstruction network are not made learnable.
> In this way, 1. The DWP module gets combined information from context vector input as well as the input-target image pair and the loss involved to update its weights. These “context-and-image” learned weights are then reshaped to the respective CNN layer size and used during testing 2. DWP allows the user to input context vectors which dynamically changes the weights of the context specific reconstruction network. This functionality transforms the reconstruction network from a context specific setting to a multi-context setting.
>
> Let us take a scenario wherein the context vector is a single element vector indicating multiple acceleration factor values, fixed studies and fixed mask pattern.  We train the model for two settings.
> Setting 1: Cardiac, Cartesian, multiple acceleration factors
> Setting 2: Cardiac, Gaussian, multiple acceleration factors
>
> We now have two sets of weights for the two settings. The learned weights for setting 1 could be very different from those of setting 2 for a given acceleration factor.  So a mapping between the context vector and input image is established.
> In the higher dimensional parameter space (say three dimensional), the context vector for T1 MRI, Cartesian mask and 4x acceleration, enumerated accordingly,  requires T1 MRI  image as undersampled input obtained from 4x acceleration and Cartesian mask pattern.
> We thank the reviewer for highlighting this point and giving us an opportunity to explain the network weight update schema in the network (i.e the back propagation part). We will add these details in the proposed architecture section (Section 3.2) of the paper as follows.
>
> During training, the loss calculated between the predicted reconstructed image and the fully sampled target image is back propagated through the reconstruction module to the DWP layers and the weights of the DWP block are learned. The weights of the CNN layers of the MRI reconstruction network are not made learnable. As a result, the weights of the DWP module are updated based on context vector input and the target-predicted image loss. The learned weights are then reshaped to the respective CNN layer size and copied during testing. DWP allows the user to input context vectors which dynamically changes the weights of the context specific reconstruction network. This functionality transforms the reconstruction network from a context specific setting to a multi-context setting.

---

### Official Review · AnonReviewer1 · 2020-03-14
**A well-framed presentation of dynamic weight prediction for improved generalization in MR image reconstruction**

**Rating:** 3
**Confidence:** 4

**Summary:**

The authors present a dynamic weight prediction module that allows reconstruction models to generalize better. This is done by training the weight prediction module is conditioned on the acquisition context vector. This means the weights are modulated by the context showing improved generalisability to unseen contexts.

**Strengths:**

The objective is well motivated and described. Generalization is very important in medical imaging and there is a clear need to handle out-of-distribution samples. The method is detailed to an appropriate level for understanding and implementation.

The experimental results are extensive, detailed and cover a range parameters space on two appropriate datasets. The results look convincing and mostly well presented.

**Weaknesses:**

The results could be presented better. Tables 1, 2 and 3 have very small font and the relationships between parameters and output is hard to gauge from reading. It would be greatly improved by visualizing these relationships in plots.

The results are impressive but would be improved with further analysis to check significance of improvement against baseline methods.

**Detailed Comments:**

Minor issues:

- It is formatted using the 2019 template (not 2020)
- 'cardiac' and 'brain' should not be capitalized
- There is too much spacing between letters in 'CNN' in equations

**Justification Of Rating:**

I believe this is a proper contribution to the field of generalizability of medical image reconstruction. The method is based on dynamic weight prediction that modulates the parameters of the reconstruction network by conditioning them on a given context encoding. The results are consistent and comparable/better than related work.

The drawbacks are mainly to do with presentation (tables are small and hard to gauge). And the impact would be improved by checking difference to baselines for significance.

**Paper Type:**

methodological development

**Questions To Address In The Rebuttal:**

1) Are the differences to CSM and JCM statistically significant? This applies to the tables and figure 5
2) How do you compensate for overfitting of the DWP?
3) How different is the DWP compared to the original paper?
4) Unseen Acceleration Factors: The current experiments are trained in range 2x - 8x and tested in range 2.4x to 7.6x. How do you expect the model to perform when the unseen factors are not within the trained range? I.e. train on 2x - 4x test on 6x.

**Special Issue:**

no

---

> ### Author Response · Authors · 2020-03-25
> **Statistical significance, overfitting,  more unseen cases**
>
> 1) Are the differences to CSM and JCM statistically significant? This applies to the tables and figure 5
> ============================================================================================
>
> CSMs give the best possible performance. Outperforming CSM is not the aim of our approach. Hence we have not performed statistical significance tests over CSM. On the other hand, statistical significance over JCM is important in every context. Wilcoxon signed-rank test
> with an alpha of 0.05 is used to assess statistical significance. We chose two such contexts and ran the test and we have obtained p-values less than 0.05.
>
> 2) How do you compensate for overfitting of the DWP?
> ============================================================================================
>
> We thank the reviewers for raising this very important limitation inherent to deep learning models and providing us the opportunity to illustrate this to add to the strengths of the paper.
>
> Context-specific models perform well in their own contexts and tend to perform poorly when the intended test scenario is not representative of the training data as is always the case in a realistic setting. This problem is called covariate shift due to overfitting.
> The ultimate aim of DWP is to limit the overfitting problem inherent in the CSMs, by providing flexibility and generalizability to multiple contexts. . The joint context model on the other hand is able to generalise well (reduce overfitting) but with some compromise in image quality. For instance, if we take our experiment on unseen cases we are able to show that the compromise on image quality can be minimised using our approach.
> We will mention the following lines in our paper about overfitting in Results and Discussion, Section 4.3 for unseen cases.
>
> The aim of DWP is to limit the overfitting problem inherent in the CSMs, by providing flexibility and generalizability to multiple contexts. . The joint context model on the other hand is able to generalise well (reduce overfitting) but with some compromise in image quality which can be minimised using our approach.
>
> 3) How different is the DWP compared to the original paper?
> ============================================================================================
>
> While in both the original paper and ours, the DWP has fully connected layers, the two are not exactly the same. This is because the base network is not the same in either case. In the original paper, the base network is a fully convolutional network whereas in ours, the reconstruction network is a deep cascade network which has alternating CNNs and data fidelity units, the DWP is designed in accordance with the DC-CNN architecture.
>
> 4) Unseen Acceleration Factors: The current experiments are trained in range 2x - 8x and tested in range 2.4x to 7.6x. How do you expect the model to perform when the unseen factors are not within the trained range? I.e. train on 2x - 4x test on 6x.
> ============================================================================================
>
> We currently do not have the results for training on 2x - 4x and testing on 6x. But we have the test results for 9x and 10x for the unseen experiment mentioned in Table 3. In this experiment, the training was done with the cardiac dataset with fixed Gaussian undersampling pattern and varying acceleration factors  - 2x, 3.3x, 4x, 5x, 8x. In Table 3, we have shown results for unseen factors - 4.8x, 5.2x, 6.0x, 6.4x, 6.8x, 7.2x and 7.6x.  To address the reviewers question, we have chosen 9x and 10x factors outside the 2x - 8x range and shown the results below.
>
> ℽ		           JCM 			               MAC-ReconNet		                           CSM
> 	             PSNR/SSIM			         PSNR/SSIM			                    PSNR/SSIM
> -----------------------------------------------------------------------------------------------------------------------------------------
> 9      29.93 +/- 3.7 / 0.8652 +/- 0.06      30.94 +/- 3.7 / 0.8849 +/- 0.06         31.06 +/- 3.5 / 0.885 +/- 0.06
> 10    29.1 +/- 3.5 /  0.8458 +/- 0.07       30.09 +/- 3.6 / 0.8658 +/- 0.07       30.32 +/- 3.5 / 0.8697 +/- 0.06
>
> These results demonstrate the generalizability of the network to unseen cases outside the parameter range.
>
>
> 5) It is formatted using the 2019 template (not 2020) 'cardiac' and 'brain' should not be capitalized
> There is too much spacing between letters in 'CNN' in equations
> ============================================================================================
>
> We will make these corrections in the paper.

---

### Official Review · AnonReviewer4 · 2020-03-14
**Addresses an important problem**

**Rating:** 4
**Confidence:** 5
**Recommendation:** Best Paper Award, Oral

**Summary:**

This paper addresses the problem of developing a deep-learning reconstruction method that is flexible enough to handle multiple acquisition contexts.  This is achieved using a reconstruction module and a dynamic weight prediction module.  Results are demonstrated in several different reconstruction settings.

**Strengths:**

Using a reconstruction module together with a dynamic weight prediction module is creative and novel.

This paper addresses an issue with very high practical relevance.  There are many imaging situations where the acquisition context is novel.  While classical reconstruction methods still work well in these situations, modern deep-learning methods generally do not.

**Weaknesses:**

The paper uses unrealistic simulations but does not discuss the limitations of these simulations.  This is an important problem to remedy, because unrealistic simulations generally have different performance characteristics than real data.  It can cause a lot of confusion and set a bad example for future research if these issues are not addressed.  The problems with the simulations include (i) the paper simulates k-space data by taking the Fourier transform of magnitude images.  Real images have phase, but these simulated images will not.  This means that the simulated data will have perfect symmetry and is much easier to reconstruct than real data would be. (ii) The simulations do not include parallel imaging, which is the standard modern approach.  (iii) There are many real k-space datasets available, I don't know why the authors didn't use some of these instead of performing unrealistic simulations.  If these issues are not fixed, they at least need to be listed as limitations and readers need to be properly warned about the interpretation of the results.

**Detailed Comments:**

Computing a zero-filled reconstruction is not ill-posed.  It is very stable, it just won't produce good results (large amounts of bias).

It's probably controversial to claim that DC-CNN is *the* state-of-the-art MRI reconstruction network.

I don't think the authors understand the meaning of "non-Cartesian".  A Gaussian sampling pattern can still be cartesian if all the sampling positions appear on the standard cartesian sampling grid.  A non-cartesian sequence has sampling positions that do not all appear on the sampling grid positions.

**Justification Of Rating:**

This is interesting and creative work.  There are some limitations, but these are fine as long as they are properly disclosed.  I think this paper is very well done, and is worthy to receive some recognition for that.

**Paper Type:**

methodological development

**Questions To Address In The Rebuttal:**

Please address the weaknesses listed above.

**Special Issue:**

yes

---

> ### Author Response · Authors · 2020-03-25
> **Adding baseline assumptions for the readers, making corrections**
>
> Reply for addressing the weaknesses mentioned
> ============================================================================================
> We preferred to explore the possibility of establishing a relationship between the acquisition context and the CNN weights using dynamic weight prediction strategy for medical imaging, MRI reconstruction in particular. Hence we ran our initial set of experiments with baseline assumptions having simple reconstruction module architecture and real data.
> We consider the comments with respect to realistic simulations using complex data and evaluations on parallel imaging very valuable. These simulations will provide a valuable insight on performance characteristics for real scenarios, help in better understanding of the model behavior and subsequently provide avenues for future exploration.
> We will clearly indicate these baseline assumptions in the Dataset and Evaluation metrics section (Section 4.1) of the paper as follows.
>
> We explore the feasibility of establishing a relationship between the acquisition context and the CNN weights of the reconstruction network using dynamic weight prediction strategy. Hence we have limited to real single coil images to demonstrate the ability of the network to multiple contexts. The k-space data used in our simulations is obtained by taking a Fourier transform of the magnitude of the images.
>
> Computing a zero-filled reconstruction is not ill-posed.  It is very stable, it just won't produce good results (large amounts of bias).
> ============================================================================================
>
> We will appropriately correct the sentence as follows.
>
> Here $F_{u}$  is the undersampled Fourier encoding matrix. For undersampled k-space measurements (M << N), this system of equations is under-determined and hence the inversion process is ill-defined. The zero filled reconstruction $x_{u}=F^{H}y$ is an aliased image due to sub-Nyquist sampling.
>
> It's probably controversial to claim that DC-CNN is *the* state-of-the-art MRI reconstruction network.
> ============================================================================================
>
> We will change the sentence as follows
> DC-CNN is one of the state-of-the-art MRI reconstruction networks.
>
> I don't think the authors understand the meaning of "non-Cartesian".  A Gaussian sampling pattern can still be Cartesian if all the sampling positions appear on the standard Cartesian sampling grid.  A non-Cartesian sequence has sampling positions that do not all appear on the sampling grid positions.
> ============================================================================================
>
> Our main intent is to check if the model is generalizable across k-space sampling trajectories. Hence we have used two different sampling mask patterns in the experiment. We apologize for the mistake afore-mentioned. We will do the following modification to the sentence about  realizing multiple masks with Gaussian and Cartesian patterns in our initial experiment as follows.
>
> The context is flexible to both scenarios where Cartesian undersampling which is practical to implement and simple to reconstruct, is preferred and other kinds of undersampling (Gaussian, spiral or radial) where higher accuracy metrics are preferred (Geethanath et al., 2013).

---

### Meta-Review · Area_Chair1 · 2020-03-31
**MetaReview of Paper56 by AreaChair1**

**Rating:** 4
**Recommendation For Accepted Papers:** Oral

**Metareview:**

Generally a good paper to address the generalization issue of deep learning method. The main contribution lies in the introduction of a dynamic weight prediction (DWP) module. The only concern I have is that the authors should avoid overcliams since the investigation of anatomy and sampling patterns is limited.

**Paper Type:**

methodological development

**Special Issue:**

no

---

### Decision · Program_Chairs · 2020-04-11

Accept